# Energy Logistic Regression and Survival Model: Case Study of Russian Exports

**DOI:** 10.3390/ijerph20010885

**Published:** 2023-01-03

**Authors:** Karel Malec, Socrates Kraido Majune, Elena Kuzmenko, Joseph Phiri, Rahab Liz Masese Nyamoita, Seth Nana Kwame Appiah-Kubi, Mansoor Maitah, Luboš Smutka, Zdeňka Gebeltová, Karel Tomšík, Sylvie Kobzev Kotásková, Jiří Marušiak

**Affiliations:** 1Department of Economics, Faculty of Economics and Management, Czech University of Life Sciences, 165 00 Prague, Czech Republic; 2School of Economics, University of Nairobi, Nairobi 30197-00100, Kenya; 3Department of Trade and Finance, Faculty of Economics and Management, Czech University of Life Sciences, 165 00 Prague, Czech Republic; 4Department of Humanities, Faculty of Economics and Management, Czech University of Life Sciences, 165 00 Prague, Czech Republic

**Keywords:** non-renewable energy exports, discrete-time model, survival analysis, Russia, environmental sustainability

## Abstract

The importance of environmental sustainability is becoming more and more obvious, so the rationale behind long-term usage of solely non-renewable energy sources appeared questionable. This study aims to identify, using Kaplan-Meier survival analysis and logistic regressions, the main determinants that affect the duration of Russian non-renewable energy exports to different regions of the world. Data were retrieved from the databanks of the World Development Indicators (WDI), World Integrated Trade Solution (WITS), and the French Centre for Prospective studies and International Information (CEPII). The obtained results point to the fact that approximately 52% of energy products survive beyond their first year of trading, nearly 38% survive beyond the second year, and almost 18% survive to the twelfth year. The survival of Russian non-renewable energy exports differs depending on the region, and the affecting factors are of different importance. The duration of Russian non-renewable energy exports is significantly linked to Russia’s GDP, Total export, and Initial export values. A decline in Russia’s GDP by 1% is associated with an increasing probability of a spell ending by 2.9% on average, in turn growing Total export and Initial export values positively linked with the duration of non-renewable energy exports from Russia. These findings may have practical relevance for strategic actions aimed at approaching both energy security and environmental sustainability.

## 1. Introduction

Rapidly changing circumstances all over the globe raised the importance of energy security to a new level of concern [1]. Since Russia is the world’s leading exporter of natural gas, the second-largest crude oil and condensates-exporting country, and the third-largest coal-exporting country as of 2021 [2], a better understanding of existing export patterns in Russia becomes crucial in terms of long-term planning for almost any economy in the world.

Figure 1 represents key energy exports from Russia to all partners as of 2021. As it can be seen, the OECD Europe obtained most of the crude oil and natural gas exports from Russia, while countries in Asia and the Oceania obtained most Russian coal exports. The ratios and numbers provided by Figure 1 depict the exporting pattern from the perspective of Russia, i.e., pointing to the composition of key Russian partners.

The importance of energy exports for the Russian economy was and remains very high (see Figure 2); however, to comprehend their importance for partners from the rest of the world, one needs to look at the sources of energy supply in these regions. It is a widely accepted fact that the resilience of the energy supply in any country may be in danger if a large proportion of energy imports is associated with relatively few external partners.

According to Eurostat data, as of 2020, energy products imported by the EU were mainly represented by the petroleum products, including their main component—crude oil. These products accounted for almost 68% of energy imports into the EU, followed by natural gas (27%) and solid fossil fuels (5%). Almost 43% of the EU’s imports of natural gas were from Russia, 21%—from Norway, 8%—from Algeria, and 5%—from Qatar. At the same time, more than half of solid fuel (mainly coal) imports were delivered by Russia (54%), the USA (16%), and Australia (14%) [3].

Establishing trade relationships is likely with factors such as sharing of a common language and culture, proximity, different factor endowment, and use of a common currency. On the other hand, the survival of trade relationships is fragile as most relationships do not survive beyond the first year [4,5]. While traditional theories of trade, such as absolute advantage, explain trade creation, they do not explain the duration of trade as it is assumed once a trade partnership is established, it will be long lasting. Establishing longer durations of trade for current trade flows is crucial as it is a critical factor in determining export growth strategy to be undertaken by a country [6,7].

Export survival is the number of uninterrupted years or months that trading partners exchange products of non-zero value [8,9]. Analysis into the subject was first introduced by Besedeš and Prusa [5,9], after which extensive empirical research has been carried out using macro data [10,11,12,13,14,15] and firm-level data [16,17,18,19]. They confirm that most trade relations between countries are interrupted after a short period of time, with some being restored after a while, while others die completely.

Various theoretical models, such as the search-cost theory by Rauch and Watson [20], product cycle theory [21] and product-switching model [22], attempt to explain the duration of trade. The search-cost theory indicates that a buyer from a developed country searches for a supplier from a developing country and begin trading. Small quantities are ordered at first with the increase in search cost, and the trade relationship is extended if the supplier is deemed reliable. Additional studies identify sunk and fixed costs as a determining factor on whether to export and for how long [7,12,17,23].

The product cycle theory [21] establishes that innovation is a key factor in a product’s transition duration in the market. It directs that a product is first produced in a developed country by highly skilled workers and after maturing, the country starts exporting the commodity to other developed countries due to an increase in foreign demand. When the product reaches the standardization stage, production is moved to a low-income country due to a comparative advantage (low wage costs and mass production). Production in the developing country creates local demand and still exports the commodity back to the developed country. Bernard et al. [22] developed the product switching theory, which envisages that firms start exporting new products to foreign markets or drop their old ones due to demand. Existing products with low demand are dropped and new products are launched due to the potential of earning higher revenues. The product switching accounts for how long a firm will be able to trade in a foreign market. Stirbat et al. [18] and Türkcan and Saygili [12] note that product and market differentiation reduce the hazard rate (this term and “export survival rate” are often used in academic literature interchangeably).

The seminal study of Besedeš and Prusa [5] empirically establishes that the duration of trade relationships is short and that imports into the US will mostly fail within the first year. In their subsequent work [10], they highlight the importance of product differentiation in extending export survival. Extended studies employing macro trade data [11,12,13,14,15] and firm-level data [17,18,19,20] have illustrated that the median period of exporting is low, and the hazard rate decreases with duration.

The trajectory of macro-based duration studies has been at a panel level [8,23,24,25] while some [6,7,11] focus on specific countries, including Kenya, Germany, and Spain, respectively. Another development has been on specific factors. For instance, the duration of agricultural and food products [26,27]. Wang, Zhu, and Wang [28] used Chinese firm-level data to establish the effects of oil shocks on export duration. They found that the duration of energy intensive industries was more affected by oil shocks than non-energy intensive industries. This is the closest study to ours given the coverage of energy intensive industries.

Other factors have been identified as determinants of export survival, besides oil shocks, namely macroeconomic factors such as GDP; exchange rate and financial development [8,14,18]; product characteristics such as type, total value, and initial export value [10,12,29]; and gravity factors such as distance, time zone differences, common borders, and trade agreements [13,15,27].

Building on the above literature, we identify some stylised facts, which we use to benchmark with our results.

1.The duration of exporting is short.

Evidence from the US indicates that the median years of imports is between 2–4 years and most trade relations do not last beyond 1 year [5,13]. The evidence is also consistent in other countries, such as Turkey, where the average length of an export relationship is 3.25 years [11]; in Germany, the median duration for importing is 2 years [11]; and most Colombian firms stop exporting to the US after the first year of entering the market [30].

2.The hazard rate decreases with duration.

An analysis of US import data from 1989–2001 using 10-digit Harmonized System (HS) dataset by Besedeš and Prusa [5] indicated that the hazard rate is 33% in the first year, 30% between the first and fifth year, and 7-12% for the remainder of the period under study. For Latin American countries, an analysis using 6-digit HS data between 1975 and 2005 indicates that the hazard rate of exporting decreases by 15% between the first and second year, a further decrease of 7% between the second and third year, a 4% decrease between the third and fourth year, and a less than 2% decrease starting from the sixth year [31]. As a result, policy implications are crucial during the first years when risk of failure is high.

3.Varied factors determine export duration.

An increase in GDP leads to a reduction in the hazard rate. An investigation on trade relationships from 1995–2004 from 96 countries indicated that the hazard rate decreases between 7% and 3% when the GDP of an exporting country doubles [8]. This is also illustrated by longer trade durations between developed countries compared to developed and developing countries [10,18,23]. An analysis on the extent to which product differentiation affects trade survival of US imports by Besedeš and Prusa [10] revealed that differentiated commodities have a higher survival rate compared to homogenous ones. Díaz-Mora et al. [6] established that firms that import and export intermediary goods simultaneously have higher survival rates.

A study in Turkey on the influence of economic integration agreements on the export of machinery between 1998–2013 found that trade survival increases for relationships that were already in existence before initiation into the economic integration agreements. This sentiment was also shared by Besedeš, et al. [32]. Besedeš and Blyde [31] also established that countries in a free trade area had a higher rate of export survival compared to those that are not in one. Distance, which is a representative of search cost, negatively affects export survival, thus increasing the hazard rate. Proximity reduces search costs, and thus promotes longer durations of trade. Empirical evidence from the study of durations and export survival of coffee exports into the EU using panel data from 1988–2013 indicates that longer distance between trading partners increases the hazard rate [27]. A similar observation was made for Argentine exports [18], US imports [13], and Lao PDR [19].

Based on previous studies and limitations, the research questions of this study are formulated the following way:

1RQ: What are the main determinants of Russian non-renewable energy exports’ survival?

2RQ: Do these determinants differ among final exporting destinations (regions of the world)?

In what follows, we firstly introduce methods and the data used, then provide the empirical results of the conducted analysis based on Kaplan–Meier survival estimate and logistic regressions. After discussing the obtained results, the study concludes by providing the answers to the main research questions.

## 2. Materials and Methods

Survival analysis was employed to establish factors that determine the duration of non-renewable energy exports from Russia. To understand survival analysis, we start by specifying the following life-table estimator of the survival function:(1)$$\hat{S}\left(j\right)=\mathrm{Pr}\left(T>j\right)=\prod_{m=1}^{j}\left(1-\frac{d_{m}}{r_{m}}\right)=\prod_{m=1}^{j}\left(1-h_{m}\right)$$
where $\hat{S}\left(j\right)$ is the survival rate at the end of a period j, T is a spell, meaning the number of years that a commodity is exported consecutively from Russia to its trading partner. A spell lasts for a period $d_{m}$, starting at $t_{m}$ and ending at $t_{m+1}$ ($d_{m}=\left(t_{m},t_{m+1}\right)\;$ for m=1,…,j). $r_{m}$, which is the adjusted number of spells at risk of failure at the midpoint of the time interval, is presented as $r_{m}=R_{m}-\frac{d_{m}}{2}$, where $R_{m}$ is the number of relationships likely to fail at the beginning of the interval. $h_{m}$ is the hazard rate, which indicates the failure of a trade relationship (spell). Jenkins [31] provides a step-by-step derivation of the Equation (1).

Equation (1) only establishes the survival rate (hazard rate) of exporting a product from Russia to another country. Hence, a discrete-time duration function needs to be specified to establish the effect covariates on the probability of exports surviving, as follows:(2)$$\mathrm{Pr}\left(y_{ijt}>0|Y_{ijt}\right)=F\left[\propto_{ij}+\delta_{ij}+\lambda_{t}+Z_{it}\beta +W_{jt}\varphi +\varepsilon_{jt,i}\right]$$
where $y_{ijt}$ measures Russia’s (i) exports to country (j) at time (t). The model controls for fixed effects by including duration ($\delta_{ij}$), spells ($\propto_{ij}$), and periods ($\lambda_{t}$). ***F***(.) is an appropriate distribution function ensuring that $\mathrm{Pr}\left(y_{ijt}>0|Y_{ijt}\right)$ ranges between 0 and 1 for all *i, j, t.* $Z_{it}$ is a vector containing product-specific characteristics—initial export value, lagged duration, and total export value, and Russia’s GDP that had a baring on its export survival. To proxy for experience, we use the initial value of export at the start of an export spell (the period a product is exported to a specific destination) and the lagged duration (the duration a previous spell lasted).

To account for the effect of experience on export survival, the total value of the exports of a product is also included. Russia’s GDP is supposed to account for the effect of the domestic production capacity on survival. $W_{jt}\;i$s a vector containing destination-specific factors, namely, gravity factors (time zone differences, distance, common border, and Regional Trade Agreements (RTA), and macro-economic indicators (importer’s GDP, exchange rate, and financial development, which measures that performance of the private sector through their access to credit and stock as proportion of GDP). RTA is also included to capture the influence of trade integration on the survival of energy exports. Other variables are supposed to show how the survival of energy exports from Russia is affected by the characteristics of the destination country. β and φ are vectors containing coefficients. $\varepsilon_{jt,i}$ is the error term.

More information about our variables along with data sources is indicated in Table 1. Please note that exports data is at HS 6-digit-destination-year level ranging from 1996 to 2019. Energy products are identified by the Harmonised System (HS) codes 270111-271600 (mineral fuels, mineral oils, and products of their distillation; bituminous substances; mineral waxes; electrical energy).

This study considered three commonly used distribution functions: logit, complementary log-log (clog-log), and probit. These functions are classified within a class of discrete-time models proved by Hess and Person [14,33] to be more suitable for duration analysis than the semi-parametric continuous-time [34] proportional hazard model. The continuous-time model suffers from unobserved heterogeneity (frailty), tied spells where relationships halt simultaneously, and the assumption of restrictive proportionality, which assumes that over time, covariates have a uniform effect on the hazard rate.

Handling left and right censoring is a common problem in survival analysis. Left-censored export records are present in the data from the first year. However, we do not know when they started. Conversely, right-censored records are active in the last year in our dataset, but we cannot know whether or when they will end. There is need to correct for left censoring otherwise, we will obtain biased estimates [14]. We do this by leaving out the first year of recorded export flows, so this study considers export flows from 1996 rather than 1995. Brenton et al. [23] and Hess and Persson [14] have found right censoring to be less problematic during survival analysis, so we include trade records for the last year in our data—2019. In line with related studies, we also include a dummy variable for multiple spells, which arise when during the study period, an export relationship stops and then relapses [5,7,35].

## 3. Results

### 3.1. Data Analysis

According to Figure 3, individual export flows, the value of which lies within the interval from 0 to 50 million USD, constitute 82% of total export flows. Export flows where values exceed 400 million USD represent almost 9% of all exports. The share of exports for 50–100 million USD is 4%. Other export values intervals account for less than 2%. Figure 3 shows the distribution of export flows at product level for the entire period of our study.

Figure 4 plots the developing trend of Russian non-renewable energy exports from 1996 to 2019. As it can be seen from this figure, since 1996 till 2008 (the beginning of the World financial crisis), the total values of Russian non-renewable energy exports were growing almost exponentially. However, after the decline in export volumes, just 1 year later, i.e., since 2009, the same exponential trend recovered and was observed till 2012, when Russia became a member of the WTO.

The period of slight stagnation lasted till 2014, after which another crisis arose associated with the Euromaidan in Ukraine [36]. This crisis was followed by the referendum in Crimea (considered as disputable by several western countries) [37,38], but some authors point out that it is “a broader problem of the correlation between the principles of maintaining a country’s territorial integrity and the right of nations for self-determination” [39,40] and consequent imposition of sanctions against Russia. All these events could not but affected the capacity of all Russian exports, not just energy ones that declined by almost USD 150,000 million. However, right after 2016 an upward trend again recovered.

Figure 5 indicates that the growing values of Russian non-renewable energy export after 2016 may be associated with an intensification of trade relations with partners from East Asia and Pacific region. As Taghizadeh-Hesary [40] asserts, Asia was the largest energy consumer in 2020, so Russia may logically count on conquering a greater portion of the Asian energy market. The natural resource wealth of Russia provides other energy importer countries a promise to contribute to their economic prosperity, a hypothesis that is supported by similar scholar, pertaining to the role of energy in enabling economic prosperity [41]

Figure 6 plots the Kaplan–Meier survival function for Russian non-renewable energy exports. About 52% of energy products survive beyond their first year of trading. Approximately 38% survive beyond the second year and 18% survive to the twelfth year. The survival rate by the twenty-fourth year, the end of our data, is 15%, suggesting, thus, that the hazard rate between the twelfth year and the end of our data is stable and just a few energy exporters exit their destination markets. The mean and median periods of energy exporting in Russia are 3.8 years and 1 year, respectively. This finding is supported by the studies based on data of such countries as USA, Turkey, Germany, and Colombia [5,11,12,13,30], respectively.

Figure 7 plots the Kaplan–Meier survival function for Russia’s non-renewable energy exports by geographical region. Export survival after the first year of trading is highest in Europe (56%), followed by South Asia (55%), Central Asia (53%), Middle East and North Africa (52%), East Asia and Pacific (49%), Sub-Saharan Africa (46%), Latin America and Caribbean (43%), and North America (27%).

### 3.2. Regression analysis

We start by identifying the suitable discrete-time model among the probit, logit, and cloglog. We find that the values and signs of the estimated coefficients (hazard rates) are qualitatively similar in all three specifications. However, we used the log-likelihood values to establish a suitable model. The log-likelihood values appear at the bottom of Table 1 and the logit model has the largest value across the three models. This suggests that the logit model offers a better fit than the probit and clog-log models. Therefore, we interpret logit results for the rest of the paper.

Table 2 displays the average marginal effects of the logit regression for non-renewable energy products by geographical regions.

## 4. Discussion

The first model indicates that every 1% increase in *Russia’s GDP* is linked, on average, to a 2.9% decrease in the probability of a spell ending, i.e., to a rise in the survival probability of non-renewable energy exports from Russia to the rest of the world by 2.9%. A percentage increase in an importer’s GDP (all groups of importers are considered by this model together) is associated with a decrease in the probability of a spell ending, on average, by 0.6%, which also indicates about an improvement of the survival of Russian non-renewable energy exports; however, at an almost 5-fold lower extent, if compared with the link to Russia’s GDP. These findings (the effect of GDP on the survival of exports) are consistent with the studies of Stirbat, Record, & Nghardsaysone [19]; Besedeš & Prusa [10]; and Brenton, Saborowski, & von Uexkull [15]. At the same time, it should be noted that the strongest effect on the survival of Russian non-renewable energy exports is exerted by a *Distance* variable, which proxies the cost of exporting. Every 1% growth in *Distance* is linked to an increase in the probability of a spell ending by 4.1%. Only this result of the first model points to a suppression of export survival or to an increase in the hazard rate. Dreyer and Anders [27]; Albornoz, Hallak, & Fanelli [18]; Besedeš [13]; and Stirbat, Record, & Nghardsaysone [19] came to a similar conclusion: proximity reduces additional costs, and thus promotes longer durations of trade. All the three above-mentioned coefficients are statistically significant. In addition, the following six coefficients also appeared to be statistically significant, and all of them are positively linked with the survival of Russian non-renewable energy exports: Common border (its existence decreases the probability of a spell ending on average by 3.4%, which supports the finding regarding *Distance* variable discussed above), partner’s real Exchange rate (its increase by 1% is associated with a decrease in the probability of a spell ending on average by 0.5%), Regional Trade Agreement (RTA) (its existence is linked to a decrease in the probability of a spell ending on average by 3.4%), Initial export value at the beginning of a spell (having increased by 1% the probability of a spell ending declines by 1.5%), Lagged duration (when the previous spell of the same export relationship lasts for 1 year longer, the probability of a current spell ending decreases on average by 0.7%), and Total export value (its increase by 1% is linked to a decrease in the probability of a spell ending on average by 1.0%). These results are in harmony with the findings of Besedeš [13]; Brenton, Saborowski, and von Uexkull [15]; Besedeš and Blyde [31]; and Dreyer and Anders [27].

The second model focuses on the analysis of East Asia and the Pacific region. In this case, *GDP* of partners from that region appeared to be insignificant in explaining the survival of Russian non-renewable energy exports to their destinations, as well as Distance, Time Zone Differences, Common border, financial development, RTA, or Lagged duration (more details about these variables are given in Table 1). Only four out of 11 resulting coefficients appeared statistically significant: Russia’s GDP (at *p* < 0.05 level), Exchange rate (at *p* < 0.10 level), Initial export value (at *p* < 0.01 level), and Total export value (at *p* < 0.10 level). Every 1% growth in these indicators is associated with a decrease in the probability of a spell ending on average by 10.1%, 1.2%, 2.8%, and 1.0%, respectively. As it can be seen, the strongest relationship is found between the survival of non-renewable energy exports from Russia to this region and the level of *Russia’s GDP*.

The third model results reveal the relationship between the analysed variables in Europe. Again, we will focus on the statistically significant results only. The highest coefficient in this model belongs to the indicator of financial development (however significant just at the *p* < 0.10 level)—its 1%-point growth is linked to an increased hazard rate growing probability of a spell ending, with a decrease in the survival probability by 3.3%. Time zone difference also appeared significant at the same *p* < 0.10 level. However, it suggests that every additional hour between local time in Russia’s capital (Moscow) and partner’s capital leads to an increase of the probability of a spell ending on average by 2.9%, meaning that the increasing time difference negatively affects survival of energy exports. In contrast, Initial export value and Total export value, when increasing by 1%, put downward pressure on the probability of a spell ending by 1.3% and 1.2%, respectively. Both parameters are statistically significant at *p* < 0.01 level and point to a rise in the survival of energy exports from Russia to Europe. Increase in Lagged duration, or the growing number of years the previous spell of the same export relationship lasted, is also positively linked to the survival of non-renewable energy exports.

The fourth model concentrates on Central Asia. There are five statistically significant parameters and four of them also point to a positive link between their growth (except for financial development) and the survival of non-renewable energy exports from Russia. Growing Exchange rate is linked with a decline in the probability of a spell ending by 1.1%, *Initial export value*—by 1.8%, Lagged duration—by 1.4%, and Total export value—by 0.8%. A 1 p.p. growth in financial development is associated with a rise in the probability of a spell ending on average by 4.1%, which is the strongest link among all the rest studied indicators for Central Asia.

The fifth model provides results that are very similar to the previous model in terms of the positive connections of all the significant indicators to the survival of non-renewable energy exports from Russia. In the case of Latin America and Caribbean, the strongest link belongs to Russia’s GDP—its 1% growth is associated with a decrease in the probability of a spell ending, on average by 15.4% (significant at p < 0.01 level). It is worth to note that this link is the strongest one among all the others in all nine models. Importer’s GDP is another significant (at *p* < 0.01 level) indicator in this model, and its growth also connected to a reduction in the probability of a spell ending, but at a lower than in case of the previous indicator extent—on average by 3.1%. The last two coefficients, both approximately equal to 1.5, reveal the same link between exports survival and Initial export value and Total export value.

The sixth model, which focuses on the Middle East and North Africa, provides just two significant parameters, 1.8 and 2.0, belonging to Initial export and Total export values, respectively. Both indicators, when growing, are linked to the increasing survival of non-renewable energy exports from Russia.

In case of North America (7th model), just one variable, the value of Total export, is statistically significant in explaining the survival of non-renewable energy exports from Russia: every 1% increase in the value of *Total export* is linked to a decrease in the probability of a spell ending on average by 2.9%.

The 8th model identifies determinants of energy fuel exports from Russia in South Asia. Among them, the following three can be listed—Exchange rate, Lagged duration, and Total export value. All their growing values are negatively linked to the duration of non-renewable energy exports from Russia, i.e., the probability of a spell ending rises by 5.4%, 2.6%, and 1.9%, respectively.

The last, 9th model investigates Sub-Saharan Africa, for which just two indicators are statistically significant—*Importer’s GDP* and the value of *Total export*. Increasing values of *Importer’s GDP* and *Total export* are associated with increasing survival of non-renewable energy exports from Russia, i.e., the probability of a spell ending declines, on average, by 3.9% and 2.7%, respectively.

From the empirical reviews of the models above, it becomes obvious that the main determinants, or factors, which affect the survival of Russian non-renewable energy exports in different regions of the world most frequently, are as follows: Total export value and Initial export value. Their impact in all cases is positive (except for South Asia, where the growth in Total export value is linked to a rise in the probability of a spell ending within the analyzed time frame). All other indicators affect the survival of Russian non-renewable energy exports in different regions of the world differently. It can be explained by the fact that OECD Europe receives most of Russia’s crude oil and natural gas exports, while countries in Asia and the Oceania region receive most of Russia’s coal exports [2]. However, the strongest link between the survival of Russian non-renewable energy exports and all the other studied indicators belongs to Russia’s GDP, which was registered for the groups of partners representing the Latin America and Caribbean region along with East Asia and the Pacific region.

Based on the obtained results, we can answer the research questions stated in the beginning of the study:

1RQ: What are the main determinants of Russian non-renewable energy exports’ survival? The conducted analysis revealed the following ones: Russia’s GDP, Total export value, and Initial export value.

2RQ: Do these determinants differ among final exporting destinations (regions of the world)? Yes, they do. Russia’s GDP is statistically significant in explaining the survival of Russian energy exports in Latin America and the Caribbean, as well as in East Asia and the Pacific region; however, for Europe, Central Asia, the Middle East and North Africa, North America, South Asia, and Sub-Saharan Africa, changes in Russia’s GDP seems to be unimportant. In turn, Total export value appears to be a major factor explaining the survival of Russian energy exports in all regions all over the world thanks to the Russian global energy market share; however, initial export value played an important role in all studied regions, with a limited impact in North America, South Asia, and Sub-Saharan Africa.

## 5. Conclusions

All statistically significant links revealed by the models (35 in total) lie within the interval from 0.5% to 15.4% of a change in the probability of a spell ending; however, most (32 out 35) are below 5.0%. From the perspective of the probability theory, these effects can be neglected. Thus, just two of the revealed effects can be considered as reliable in terms of future predictions of the survival of energy exports from Russia, and both are connected to the GDP of Russia. A decline in Russia’s GDP is statistically significantly linked to a growth in the probability of a spell ending, which points to a downward trend in the survival of non-renewable energy exports from Russia. At the same time, the constructed Kaplan-Meier survival function for Russian non-renewable energy exports disclosed the fact that about 52% of energy products survive beyond their first year of trading, approximately 38% survive beyond the second year, and nearly 18% of energy products survive to the twelfth year (the mean and median periods of Russian energy exports are 3.8 years and 1 year, respectively). These findings may have practical relevance for strategic actions aimed at approaching both energy security and environmental sustainability in different regions of the world; these could be through instituting through favorable RTAs between Russian and its partner counties, a stable financial and forex market across the globe, and diverting a huge proportion of Russian energy exports to countries with a high GDP and sustained economic growth, as this will proliferate the number of imports from Russia to importer countries and increase the survival probability of all its energy exports both renewable and non-renewable energy exports.

## Figures and Tables

**Figure 1 ijerph-20-00885-f001:**
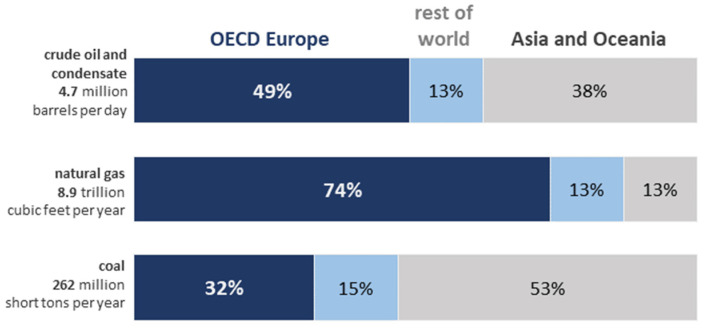
Key energy exports from Russia (2021). Source: [2].

**Figure 2 ijerph-20-00885-f002:**
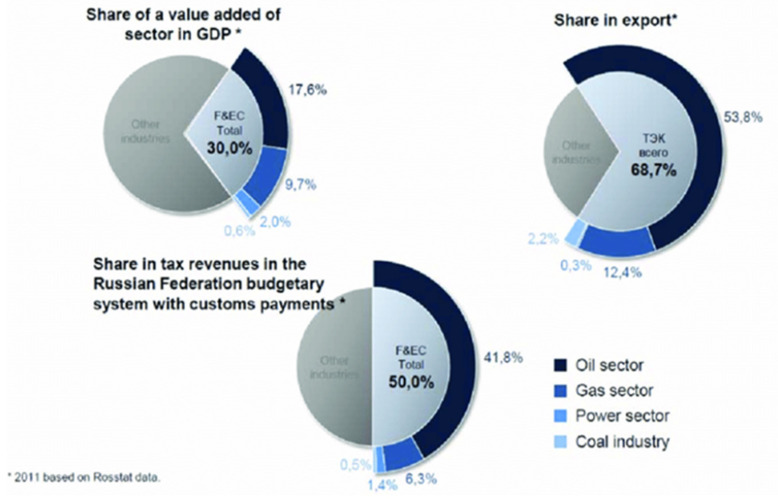
The importance of the Energy Sector for GDP, Export, and Budget Revenues of Russia. Source: [4].

**Figure 3 ijerph-20-00885-f003:**
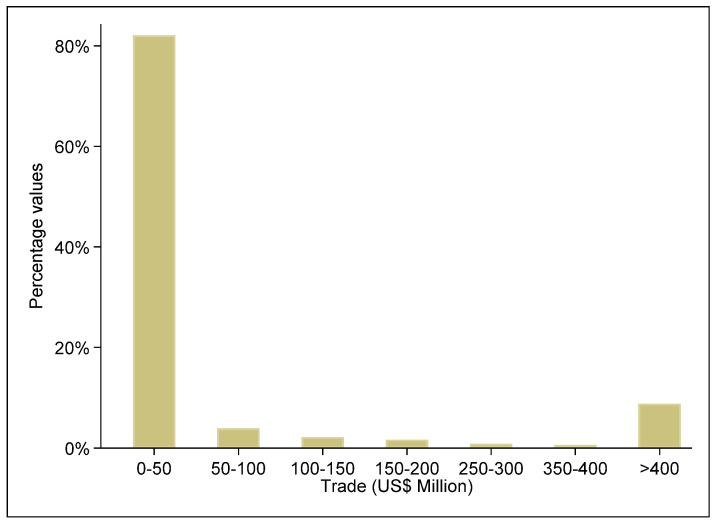
Distribution of non-renewable energy export values by percentage. Source: Authors’ elaboration based on data retrieved from CEPII database.

**Figure 4 ijerph-20-00885-f004:**
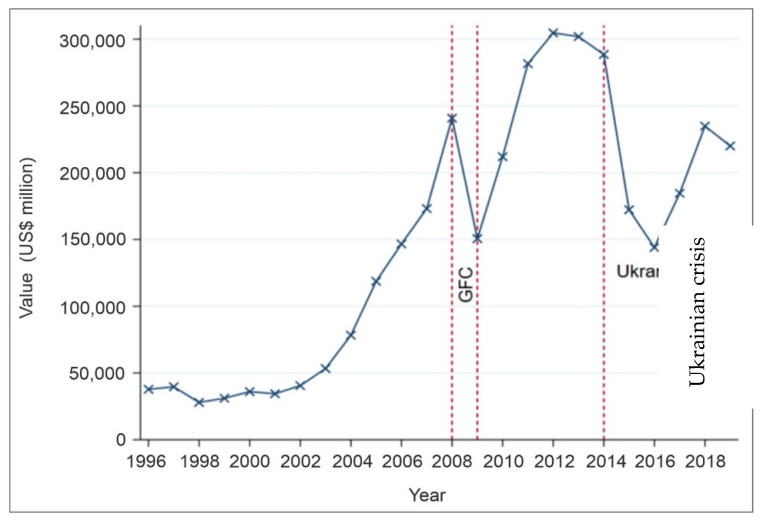
Russian non-renewable energy export values (1996–2019). Note: GFC is Global Financial Crisis that happened in 2008/9, while abbreviation Ukran is the country Ukraine with the event in the highlighted period Ukrainian crisis been the annexation of Crimea. Source: Authors’ elaboration based on data retrieved from CEPII database.

**Figure 5 ijerph-20-00885-f005:**
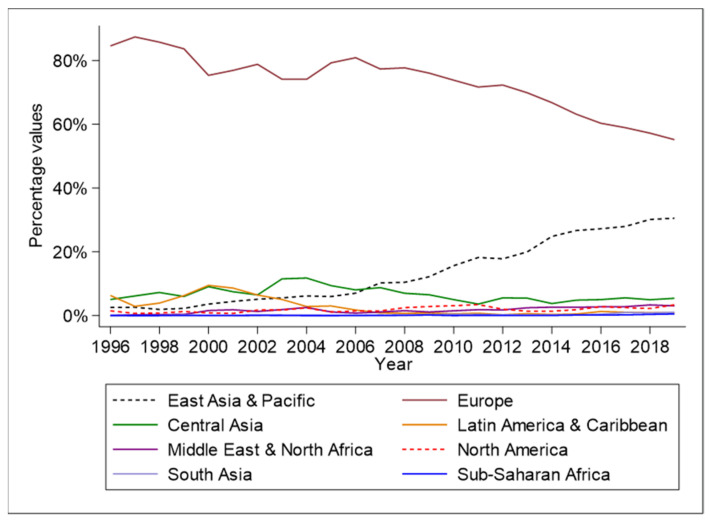
Russian non-renewable energy export values by region (1996–2019). Source: Authors’ elaboration based on data retrieved from CEPII database.

**Figure 6 ijerph-20-00885-f006:**
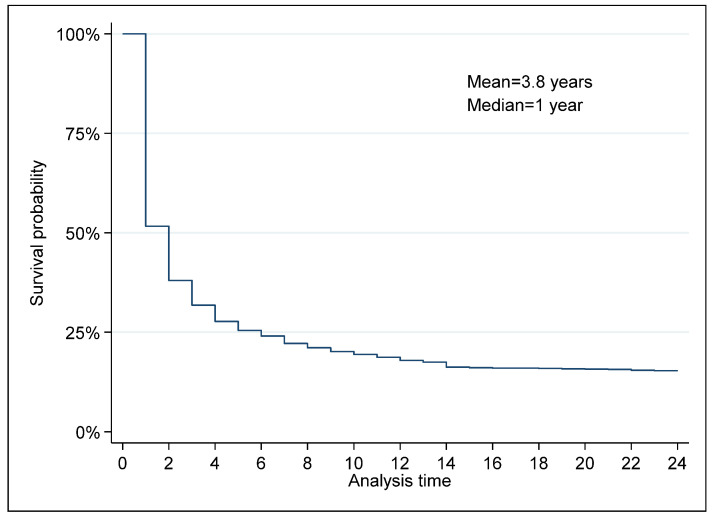
Kaplan–Meier survival estimate for Russian non-renewable energy exports. Source: Authors’ elaboration based on data retrieved from CEPII database.

**Figure 7 ijerph-20-00885-f007:**
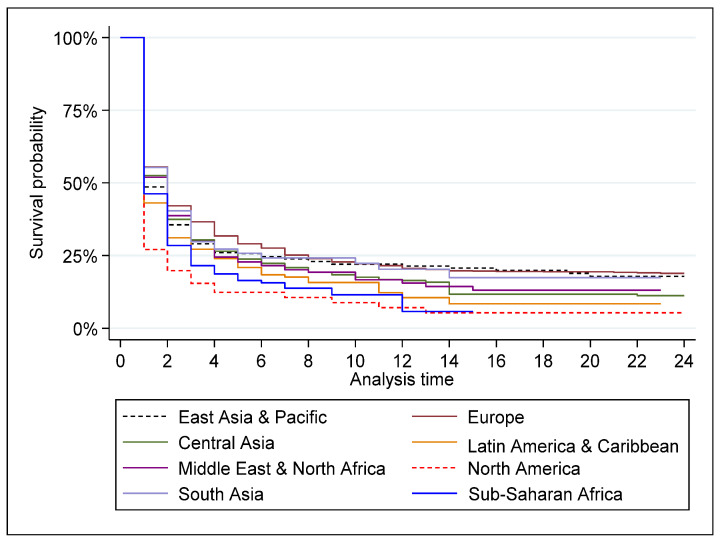
Kaplan–Meier survival estimate for Russian non-renewable energy exports by geographical region. Source: Authors’ elaboration based on data retrieved from CEPII database.

**Table 1 ijerph-20-00885-t001:** Regressions results for duration analysis with random effects.

	(1)	(2)	(3)
	Cloglog	Probit	Logit
GDP Russia	−0.1579 **	−0.1432 **	−0.2352 **
	(0.080)	(0.069)	(0.116)
GDP importer	−0.0288 **	−0.0314 ***	−0.0480 **
	(0.013)	(0.011)	(0.019)
Distance	0.2063 ***	0.2050 ***	0.3423 ***
	(0.062)	(0.056)	(0.093)
Time Zone Difference	0.0053	0.0037	0.0041
	(0.019)	(0.017)	(0.029)
Common border	−0.1947 ***	−0.1600 ***	−0.2739 ***
	(0.066)	(0.058)	(0.096)
Financial Development	−0.0267	−0.0317	−0.0520
	(0.034)	(0.029)	(0.049)
Exchange rate	−0.0296 ***	−0.0249 ***	−0.0432 ***
	(0.010)	(0.009)	(0.015)
RTA	−0.1711 **	−0.1627 ***	−0.2671 ***
	(0.066)	(0.058)	(0.097)
Initial export value	−0.0861 ***	−0.0709 ***	−0.1214 ***
	(0.008)	(0.007)	(0.011)
Lagged duration	−0.0640 ***	−0.0287 ***	−0.0557 ***
	(0.017)	(0.011)	(0.020)
Total export value	−0.0463 ***	−0.0507 ***	−0.0820 ***
	(0.008)	(0.007)	(0.013)
Duration dummies	Yes	Yes	Yes
Spell dummies	Yes	Yes	Yes
Year dummies	Yes	Yes	Yes
Observations	12,393	12,393	12,393
Spells	3652	3652	3652
Trade relations	1984	1984	1984
Log-likelihood	−4766	−4762	−4759

Note: *, **, and *** indicate statistical significance at the 10%, 5%, and 1% level, respectively. The dependent variable is a binary variable that equals one if an export spell is ended and zero otherwise. Source: Authors’ elaboration based on data retrieved from CEPII database.

**Table 2 ijerph-20-00885-t002:** Logit regression marginal effects.

	(1)	(2)	(3)	(4)	(5)	(6)	(7)	(8)	(9)
	Total	East Asia and Pacific	Europe	Central Asia	Latin America and Caribbean	Middle East and North Africa	North America	South Asia	Sub-Saharan Africa
GDP Russia	−0.0292 **	−0.1011 **	−0.0074	0.0530	−0.1538 **	−0.1835	−0.8361	0.0833	0.0250
	(0.014)	(0.048)	(0.021)	(0.043)	(0.078)	(0.122)	(1.989)	(0.104)	(0.143)
GDP importer	−0.0059 **	−0.0087	−0.0028	−0.0020	−0.0307 ***	−0.0178	1.3618	0.0138	−0.0393 **
	(0.002)	(0.006)	(0.004)	(0.011)	(0.011)	(0.019)	(3.050)	(0.044)	(0.018)
Distance	0.0411 ***	0.0476	0.0451	0.0737	0.0677	−0.0632	−35.2179	0.3125	−0.0393
	(0.011)	(0.115)	(0.032)	(0.140)	(0.129)	(0.146)	(78.102)	(0.497)	(0.182)
Time zone difference	0.0006	0.0032	0.0285*	0.0014	−0.0082	0.0087	−10.7320	0.0258	−0.0301
	(0.004)	(0.020)	(0.017)	(0.030)	(0.016)	(0.038)	(24.637)	(0.271)	(0.023)
Common border	−0.0343 ***	−0.0834	−0.0098	0.0200	-	-	-	-	-
	(0.012)	(0.057)	(0.023)	(0.047)	-	-	-	-	-
Financial development	−0.0054	−0.0186	0.0329 *	0.0405 **	0.0165	0.0973	−0.0785	0.0300	0.0126
	(0.006)	(0.013)	(0.020)	(0.020)	(0.027)	(0.084)	(0.071)	(0.345)	(0.052)
Exchange rate	−0.0046 **	−0.0117 *	0.0004	−0.0114 *	0.0026	0.0040	2.1269	0.0540 **	−0.0084
	(0.002)	(0.006)	(0.004)	(0.006)	(0.009)	(0.009)	(6.040)	(0.026)	(0.013)
RTA	−0.0335 ***	0.0263	−0.0117	−0.0328	−0.1249	-	-	-	-
	(0.012)	(0.141)	(0.021)	(0.056)	(0.144)	-	-	-	-
Initial export value	−0.0150 ***	−0.0275 ***	−0.0130 ***	−0.0179 ***	−0.0145 ***	−0.0180 ***	−0.0086	−0.0076	−0.0085
	(0.001)	(0.005)	(0.002)	(0.004)	(0.006)	(0.007)	(0.009)	(0.009)	(0.008)
Lagged duration	−0.0070 ***	−0.0018	−0.0061 *	−0.0136 **	−0.0139	−0.0074	−0.0740	0.0261 **	−0.0225
	(0.002)	(0.009)	(0.003)	(0.006)	(0.015)	(0.021)	(0.059)	(0.011)	(0.028)
Total export value	−0.0102 ***	−0.0104 *	−0.0116 ***	−0.0077 **	−0.0159 **	−0.0197 **	−0.0278 *	0.0191 *	−0.0265 ***
	(0.002)	(0.006)	(0.002)	(0.004)	(0.007)	(0.008)	(0.015)	(0.011)	(0.009)
Observations	12,393	1132	6533	2101	478	630	160	259	369
Spells	3643	3643	3643	3643	3643	3643	3643	3643	3643
Trade relations	1978	1978	1978	1978	1978	1978	1978	1978	1978

Note: The table reports coefficients of the average marginal effects and the corresponding robust standard errors clustered at the importer-product level. *, **, and *** indicate statistical significance at the 10%, 5%, and 1% level, respectively. The dependent variable is a binary variable that equals one if an export spell is ended and zero otherwise. Source: Authors’ elaboration based on data retrieved from CEPII, WDI and WITS databases.

## Data Availability

The data used in the analysis were retrieved from WDI, WITS, and CEPII databanks.

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
