# Peer review of "Energy Logistic Regression and Survival Model: Case Study of Russian Exports"

_ijerph, 2023, doi:10.3390/ijerph20010885_

Round 1

Reviewer 1 Report

Dear Authors,

With great interest, I read your manuscript on the survival of Russian exports of energy carriers.  Its main contribution and strength is the selection of your research topic, which arrives at times when a better understanding of Russian energy trade is urgently needed.

Still, in my opinion, your paper requires major changes before being accepted for publication. In particular, in view of your method description including the data sources you used, I think it is quite likely that the scope of your study goes beyond non-renewable energy, as you considered Russian exports of electrical energy, which most likely includes electrical energy from renewables. I kindly ask you to clarify this, as the study title, research questions and other parts of your paper might require substantial revisions in view of this observation.

Please find below my specific comments and suggestions for improvements:

1.     Abstract: please provide the full names of WDI, WITS and CEPII, as these abbreviations may not be widely understood by your readers.

2.     Abstract: “, and the affecting factors are of different importance”  –  consider removing, as this phrase is quite generic and provides little information value.

3.     Page 3, line 8 from the top: I suggest replacing the word “literature” with “research”, as literature cannot be “carried out”.

4.     Page 3, line 11 from the bottom: “duration of energy intensive industries” – I suggest clarifying

5.     Page 4, line 8 from the top: please explain the abbreviation HS

6.     Page 5, line 7 from the top: please check the numbering of all references – for instance, Jenkins should be numbered as [31], not [32].

7.     Page 5, line 17 from the top: please check if the word “factors” in the phrase “Russia-specific factors” shouldn’t be singular, not plural. If there are more factors included, please provide them in the parenthesis next to the GDP.

8.     Page 5, para. 4 from the top: please provide a disambiguation of RTA

9.     Page 5, para. 4 from the top: please explain how the ‘financial development’ indicator is modeled. 

10.  Page 5, para. 6 from the top: please provide the Table A.1 – I have not found it in the manuscript. It is not available in the MDPI Submission System either.

11.  Page 5, para. 6 from the top: please explain if the category “electrical energy” includes only electrical energy generated from non-renewable sources. If that’s not the case, please revise the whole manuscript, including the title. To the best of my understanding, your data involved any type of electrical energy exports, including especially hydropower-sourced electricity, which is important in Russian electricity balance. Therefore, the references narrowing down the scope of research to “non-renewable energy” are not accurate.

12.  Page 7, Fig. 4: please explain/revise the word “Ukran” on the figure. 

13.  Page 7, Fig. 4: please provide an explanation for GFC. 

14.  Page 7, Fig. 4: please change the phrase “Ukrainian crisis” – it is not clear.

15.  Page 7: parenthesis starting in the phrase “Crimea (considered…”) has no end. Please add. 

16.  Page 7, last line of text: please add “million” next to “USD 150 000”.

17.  Page 8, last two lines of text: consider revising to avoid double mentioning of USA, e.g. “USA [4,10], Turkey [11], …” and then remove the last word (“respectively”).

18.  Page 11, the second line of discussion: please verify “a rise in the survival…” – please check if the reference to probability should not be repeated here. 

19.  Page 12, para. starting with “The third model…”, line 4: please check/revise throughout the text that the interpretation of changes in indexed indicators (%) linked with  is correct. Here, you indicate that 1 percentage point growth in Financial development Is linked to the higher probability by 3.3%. 

20.  Page 13, two last lined of discussion section: consider revising these lines, as they are generic and would fit any research paper. 

21.  Page 14, line 2 from the top: please consider replacing the word “fruitful” with a more accurate one in this context.

22.  Page 14, last sentence: please consider explaining what “strategic actions” you mean. Consider providing examples. Your further thoughts on the last sentence could be placed in a more comprehensive separate paragraph.

In your next paper version and future papers, I would also suggest numbering the lines. 

Author Response

Dear Editor/Reviewers,

We would like to thank you for your helpful comments that helped improve our manuscript. We worked on the recommendations made, and below is a commentary of actions done. Once more, thank you and we look forward to your feedback:

Still, in my opinion, your paper requires major changes before being accepted for publication. In view of your method description including the data sources you used, I think it is quite likely that the scope of your study goes beyond non-renewable energy, as you considered Russian exports of electrical energy, which most likely includes electrical energy from renewables. I kindly ask you to clarify this, as the study title, research questions and other parts of your paper might require substantial revisions in view of this observation: The context is maintained though emphasize was on non-renewable energy is it constituted a huge component of Russian energy exports; please note the topic has been revised to incorporate all components of energy produced in an exported from Russia.

  1. Abstract: please provide the full names of WDI, WITS and CEPII, as these abbreviations may not be widely understood by your readers: Abbreviations explained.
  2. Abstract: “, and the affecting factors are of different importance” – consider removing, as this phrase is quite generic and provides little information value
  3. Page 3, line 8 from the top: I suggest replacing the word “literature” with “research”, as literature cannot be “carried out”: Amended
  4. Page 3, line 11 from the bottom: “duration of energy intensive industries” – I suggest clarifying
  5. Page 4, line 8 from the top: please explain the abbreviation HS: Explained
  6. Page 5, line 7 from the top: please check the numbering of all references – for instance, Jenkins should be numbered as [31], not [32]: Corrected
  7. Page 5, line 17 from the top: please check if the word “factors” in the phrase “Russia-specific factors” shouldn’t be singular, not plural. If there are more factors included, please provide them in the parenthesis next to the GDP: Corrected
  8. Page 5, para. 4 from the top: please provide a disambiguation of RTA: Corrected
  9. Page 5, para. 4 from the top: please explain how the ‘financial development’ indicator is modeled. Explained
  10. Page 5, para. 6 from the top: please provide the Table A.1 – I have not found it in the manuscript. It is not available in the MDPI Submission System either: Relabeled corrected
  11. Page 5, para. 6 from the top: please explain if the category “electrical energy” includes only electrical energy generated from non-renewable sources. If that’s not the case, please revise the whole manuscript, including the title. To the best of my understanding, your data involved any type of electrical energy exports, including especially hydropower-sourced electricity, which is important in Russian electricity balance. Therefore, the references narrowing down the scope of research to “non-renewable energy” are not accurate: Corrected and title changed
  12. Page 7, Fig. 4: please explain/revise the word “Ukran” on the figure: Explained below figure
  13. Page 7, Fig. 4: please provide an explanation for GFC: Provided
  14. Page 7, Fig. 4: please change the phrase “Ukrainian crisis” – it is not clear: Clarified
  15. Page 7: parenthesis starting in the phrase “Crimea (considered…”) has no end. Please add: Corrected
  16. Page 7, last line of text: please add “million” next to “USD 150 000”: Corrected
  17. Page 8, last two lines of text: consider revising to avoid double mentioning of USA, e.g., “USA [4,10], Turkey [11], …” and then remove the last word (“respectively”): Corrected
  18. Page 11, the second line of discussion: please verify “a rise in the survival…” – please check if the reference to probability should not be repeated here: Corrected 
  19. Page 12, para. starting with “The third model…”, line 4: please check/revise throughout the text that the interpretation of changes in indexed indicators (%) linked with is correct. Here, you indicate that 1 percentage point growth in financial development Is linked to the higher probability by 3.3%: Corrected 
  20. Page 13, two last lined of discussion section: consider revising these lines, as they are generic and would fit any research paper.: Revised
  21. Page 14, line 2 from the top: please consider replacing the word “fruitful” with a more accurate one in this context: Corrected
  22. Page 14, last sentence: please consider explaining what “strategic actions” you mean. Consider providing examples. Your further thoughts on the last sentence could be placed in a more comprehensive separate paragraph: Explained

The submitted manuscript is interesting and topical. There are, however, some issues that must be fixed. The issue of the commodity exports from Russia, especially the non-renewable must be embedded into the context of the natural resources - growth/development nexus. Hereby, the question of the resource curse is the key concept. Do it at least cursorily and refer to the following source, which is indispensable in this regard - https://doi.org/10.3390/economies7040113 In addition, the conclusions are too short. Extend it by a comprehensive discussion of the findings. Refer also to the current situation: Corrected and some reference added

Your positive feedback on these reviews is highly appreciated

Professor. Mansoor Maitah.

Lecturer and Researcher – Department of Economics.

Faculty of Economics and Management.

Czech University of Life Sciences Prague.

[email protected]

+420224382422.

Reviewer 2 Report

The submitted manuscript is really interesting and topical. There are, however some issues that must be fixed. The issue of the commodity exports from russia, especially the nonrenewables must be embedded into the context of the natural resources - growth/development nexus. Hereby, the question of the resource curse is the key concept. Do it at least cursorily and refer to the follwing source, which is indispensable in this regard - https://doi.org/10.3390/economies7040113 In addition, the conclusions are too short. Extend it by a comprehensive discussion of the findings. Refer also to the current situation.

Author Response

(The authors gave the same response as above.)

Round 2

Reviewer 1 Report

Dear Authors,

Thank you for your reply. In the pdf file you uploaded ("ijerph-2087551-peer-review-v2.pdf") I cannot find the changes that you indicated that you have implemented. For instance, in your cover letter, you indicated that you changed the title of the paper. Still, in the resubmitted manuscript, the title remains the same.

Please resubmit the manuscript - maybe you uploaded the wrong pdf file by mistake.

I would appreciate a version in track-changes mode. Thanks in advance.

Author Response

Dear Respected sirs, you are right. We have done everything, but now we are unable to upload the last version. It is attached below and we will send it by email too.
